# The Systems Approach and Design Path of Electronic Bidding Systems Based on Blockchain Technology

De Xu [1,2] and Qing Yang [3,*]

1   Intelligent Construction and Blockchain Collaborative Innovation Research Center, Jiangsu Open University, Nanjing 210000, China
2   Zhongru Information Technology Co., Ltd., Nanjing 210000, China
3   School of Architecture and Engineering, Jiangsu Open University, Nanjing 210000, China
*   Correspondence: yangq@jsou.edu.cn

**Abstract:** The electronic tendering and bidding system has realized the digitalization, networking, and high integration of the whole process of tendering, bidding, bid evaluation, and contract, which has a wide range of applications. However, the trust degree, cooperation, and transaction efficiency of the parties involved in electronic bidding are low, and bidding fraud and collusion are forbidden repeatedly. Blockchain technology has the characteristics of decentralization, transparent transactions, traceability, non-tampering and forgery detection, and data security. This paper proposes a design path of an electronic bidding system based on blockchain technology, which aims to solve the efficiency, trust, and security of the electronic trading process. By building the underlying architecture platform of blockchain and embedding the business process of electronic bidding, this realizes the transparency, openness, and traceability during the whole process of electronic bidding. This paper uses qualitative and quantitative methods to prove the effectiveness of the system.

**Keywords:** blockchain technology; electronic bidding; system design





## 1. Introduction

The construction industry plays an important role in the development of the social economy. However, the traditional bidding method has problems such as low information transparency, information asymmetry, and an opaque transaction process, which inhibit the development of the construction industry. Compared with the traditional bidding method, the electronic bidding (E-bidding) system is an essential transaction method in the current information era [1], which consolidates the process of bidding, tendering, bid evaluation, and contract-signing as an open network, and breaks through the limitation of time and space. Additionally, since the emergence of COVID-19 through the spread of Omicron, in the context of the global scale, it is of great practical significance to study how to realize the whole process of E-bidding online and how to ensure continuous economic activities and reduce personal contact.

Although the E-bidding system is widely used at present, the following problems still exist [2]. Firstly, there is no unified standard for E-bidding systems, which leads to poor real-time collaboration within different systems. Secondly, in the E-bidding system, it is difficult to ensure the identity authentication of users and data security, which is affected by network security. Finally, the E-bidding system has difficulties in the traceability of bidding participant behaviors. Namely, unfair situations often occur in transactions, but it is difficult for regulators to collect bidding fraud evidence and achieve supervision in real-time.

To solve the above problems, blockchain technology is brought into the E-bidding system. As a decentralized distributed ledger, blockchain co-records public data in chronological sequence, generates and updates data through distributed node consensus algorithm, and employs cryptography technology to ensure that data cannot be tampered with. Naturally, the blockchain enables collaboration without the authorization of a third-party,

facilitates the construction of a highly credible transaction and supervision system with high security and reliability, and can trace all information in the transaction process to ensure transparency and fairness [3,4]. To this end, blockchain will become one of the most prevalent underlying protocols of the "Internet of Everything" and be applied in all fields of society, i.e., social governance, arbitration, auditing, smart city construction, etc. However, compared with other industries, it is the diverse and complicated transaction process that hinders the development of E-bidding in construction sectors. In addition, the application of E-bidding in the construction industry has lagged behind the manufacturing and retailing sectors [5], not to mention the adaptation of blockchain-based E-bidding. In addition, the research on blockchain-based E-bidding systems is limited to the preservation of information in each stage of bidding and does not consider how to avoid bidding fraud to maintain fair transactions. Thus, to promote further prosperity, decrease the large resource consumption, and improve the overall transaction process's efficiency and security, it is necessary to study blockchain-based E-bidding in the construction industry.

In this paper, we combine blockchain technology and an E-bidding system and propose a blockchain-based E-bidding system applied in the construction industry, which consists of a blockchain electronic transaction bidding system, a big data system, and a framework for mining evidence of bidding fraud. By virtue of a large amount of complex and frequently changing transaction information to handle, it is time-consuming and a great challenge for the E-bidding system to collect, process, and analyze the large-scale data. Hence, the introduction of "big data" technology into the blockchain-based E-bidding system will promote the interconnection and real-time sharing of information, as well as further optimize the market-based allocation of resources. In addition, bidding fraud detection is also an essential issue of concern in E-bidding. The "big data" analysis can quickly determine whether there is bidding fraud or collusion in the bidding process and provide fair digital "evidence" to assist the bidding administrative department to strengthen regulation of the entire bidding process and impose administrative penalties for violations, which is advantageous in improving the standardization, digitalization, and scientific level of bidding activities.

The main contributions of this work can be summarized as follows:

(1) This paper combines blockchain technology and an E-bidding system in the construction industry and designs a blockchain-based E-bidding framework to raise bidding efficiency and guarantee the fairness, impartiality, and transparency of transactions.

(2) On the basis of big data technology, a big data system (BDS) is designed to collect, handle, and analyze the data in the bidding process, which is convenient for maintaining transaction fairness and improving bidding efficiency.

(3) A bidding fraud evidence mining method is embedded in the big data system to mine fraud evidence and strengthen transaction supervision, which combines maximal frequent itemset mining, association rule mining, and binary support number calculation algorithms to boost operational efficiency.

The remainder of this paper is organized as follows. Section 2 offers the related work of E-bidding systems in the construction industry, application of blockchain, blockchain-based E-bidding systems in the construction industry, and big data system. Section 3 provides the proposed blockchain-based E-bidding systems. Section 4 shows the extensive experiments and results of the proposed method for electronic bidding. Section 5 presents the conclusion.

## 2. Related Work

### 2.1. Electronic Bidding System in the Construction Industry

The traditional project bidding field has gone through a long road of development under the norms of laws and regulations such as the "Tendering and Bidding Law" and the "Government Procurement Law", which have played an important role in unifying the rules of the bidding market and encouraging orderly competition in the market. The emergence and wide application of the internet is a revolution in industrial society; for the

construction industry in the field of engineering bidding, the emergence and development of electronic bidding has also redefined the ways and methods of bidding by construction market entities and has played a positive role in further promoting a free, fair, just, and honest market environment. In recent years, the government and relevant industry organizations have supported and encouraged construction units and relevant market entities to carry out electronic bidding and bidding work, which has effectively promoted the application of electronic bidding. The electronic bidding system realizes business functions such as online bidding, bidding, bid evaluation, and contract management, reduces offline transaction costs, improves work efficiency, enhances the information management capabilities of governments and participating entities, and effectively promotes the digitalization, networking, and high integration of the whole bidding process. However, there are still some problems with the current electronic tendering, resulting in the application of electronic tendering still being quite limited. First, relevant laws and regulations lag behind, there are a lack of unified norms and standards, and it is difficult to promote. Second, there are many electronic bidding platforms, which are poorly compatible with each other, and the phenomena of administrative intervention and secret operations cannot be effectively prevented. Third, the security and stability of the electronic bidding platform need to be strengthened; if the data security and stability performance is not effectively guaranteed, it is very easy to enable the leakage of commercial secrets and malicious tampering of data information. The research on these problems is of great practical significance for the application of electronic bidding in the construction industry.

### 2.2. Application of Blockchain

Blockchain, sometimes known as a distributed shared ledger, is essentially a multi-participant, cooperatively maintained, continually growing distributed database system. Blockchain technology is very well liked by businesses and has been widely used because of its anonymous, decentralized, open and transparent, and tamper-evident characteristics. In the field of finance, when blockchain peer-to-peer (P2P) technology was applied to cross-border payments [6], the remittance becomes transparent, and transaction history data was traceable, providing security assurance for both the recipient and the remitter while also considerably enhancing efficiency and speed. In addition, with the application of blockchain in medical data privacy protection [7], medical data storage and access can be recorded and remain tamper-proof, which avoids unscrupulous individuals from using this information for fraud and blackmail. Also, the untamperable nature of blockchain renders the digital proof on the chain extremely believable, which may be utilized to create a new authentication mechanism in the areas of property rights [2], notarial services [8], and social welfare [9] and to raise the management standard of public service. Motivated by the compatibility of blockchain characteristics with trade process requirements, we attempt to integrate blockchain into the E-bidding system with its advantages of distribution, anonymity, transparency, and traceability to promote the reform and progress of the E-bidding system in the construction industry.

### 2.3. Blockchain-Based E-Bidding in the Construction Industry

Since the structure and technology of blockchain effectively ensure the authenticity and traceability of information, the research on the application of E-bidding systems in the construction industry has become popular in recent years. In 2017, motivated by the dynamic grouping of several companies in the projects, Turk et al. [10] introduced the P2P nature of the relationships in blockchain technology to establish a reliable infrastructure for information management throughout all stages of the building life-cycle. To improve the data reliability and verifiability and privacy of data transmission, Tso et al. [11] applied blockchain and smart contract technology and proposed the first decentralized electronic voting and bidding systems. In 2021, Sigalov et al. [12] combined Building Information Modeling (BIM) approaches with smart contracts to achieve automated billing, which enhances timely payment and guaranteed cash flow. Compared with these approaches,

our method has higher operation efficiency and can mine bidding fraud evidence through big data technology, which will be later described in detail.

### 2.4. Big Data Technology

Big data technology has the following four characteristics: Volume, Variety, Value, and Velocity [13] when compared with traditional databases. With these advantages, after collecting and organizing the large-scale data, it is much easier and more practical to determine its potential laws and predict the development trend through intelligent analysis and data mining. This can assist people in decision-making [14], boost operational efficiency, and realize greater benefits. Therefore, there are many applications of big data technology in our daily life [15,16], such as finance [17,18], E-commerce [19], medical [20], and communication [21]. Moreover, it is data analysis that is the key point of big data technology, which usually uses data mining to acquire the diagnosis of anomalous data. In 2000, Pei et al. [22] proposed an efficient and scalable algorithm for frequent closed itemset mining with the use of a frequent pattern (FP) tree, which could provide a minimum description of abnormality. To reduce the computational complexity and memory usage, Halim et al. [23] presented a graph-based approach with storage of all relevant information to mine maximal frequent itemsets and prove its superiority. With only one access to the record of all frequent itemsets, it can significantly improve the execution efficiency of positive as well as negative association rule mining [24,25] and further increase the run-time efficiency of the whole process. Hence, we employ big data technology to assist in evaluating bidding activities, ensuring project quality, and boosting operational effectiveness while also providing reliable decision-making support for all types of transaction issues. Moreover, there is little research to study how to assist the blockchain-based E-bidding system through big data technology. Inspired by this, we integrate a big data system (BDS) into the E-bidding system in this work.

## 3. Method

### 3.1. Preliminaries of Blockchain

In this section, we introduce some preliminaries about the blockchain to which the traditional E-bidding system is adjusted.

#### 3.1.1. Definition of Blockchain

Blockchain is generally considered as a decentralized, de-trusted, distributed, shared ledger system that combines blocked data, which includes transaction information, timestamps, and hash value in a chain chronologically and cryptographically [26]. From the view of data, blockchain can be interpreted as a distributed database that cannot be passively modified or forged. From the view of technic, blockchain is a distributed ledger technology integrated with various technologies, such as asymmetric cryptography [27], P2P network [28], and smart contracts [29].

#### 3.1.2. Characteristics of Blockchain

A key characteristic of blockchain is that it is a distributed and decentralized system. While only one controller manages the completeness of data information in a centralized database [30], the term "distributed system" means that the content of transaction information can be stored and examined simultaneously by all participants, which makes it possible to maintain information integrity and trustworthiness without the need for authorization. The use of various distributed applications [31] is to achieve state change management, data storage, query validation, and control management. Therefore, blockchain has more obvious technical and management advantages compared with traditional centralized systems.

Additionally, using hash algorithms as encryption technology, the most prominent advantage of blockchain is its high level of security [32]. Since the information is all jointly owned in the blockchain, when viruses or hackers attack P2P-specific data, they cannot change or delete data at will. Secondly, a decentralized blockchain can minimize transac-

tion costs to a maximum extent while having good technical scalability and improving transaction efficiency. Finally, blockchain, due to its openness nature, can improve the transparency and fairness of transactions, ensure security, and reduce regulatory costs. Although the access rights in the blockchain vary, almost all participants can access all the transaction records and information stored by the chain blocks anytime and anywhere [33]. All the above characteristics are summarized in Table 1.

**Table 1.** The characteristics of blockchain [3,4,6–9].

| Characteristics | Description |
| --- | --- |
| Decentralization | Each node realizes information self-verification, self-transmission, and self-management. |
| Immutability | No one can modify the data without authorization once it has been written to the blockchain. |
| Security | All data on the chain are encrypted by hash operation, asymmetric encryption, private key, and other cryptographic methods. |
| Openness | All nodes in the chain can participate in the record maintenance of data. |

### 3.1.3. Categories of Blockchain

The classification of blockchain is based on the degree of network openness and can be mainly classified as public, private, and industry blockchains [34,35], which is shown in Table 2. Concretely speaking, a public blockchain is a blockchain shared by any organization or individual that can operate and be confirmed on that blockchain, and other organizations or individuals can join it; a private blockchain is one in which the blockchain is used only internally for bookkeeping activities; a consortium blockchain is one in which some nodes are controlled by pre-selected nodes.

**Table 2.** The categories of blockchain.

| Categories | Description | Scenarios | Trust Authority | Speed of Consensus |
| --- | --- | --- | --- | --- |
| Public Blockchain | Anyone can operate and be confirmed. | Virtual Cryptocurrency | 0 | Slow |
| Private Blockchain | An organization controls the write access. | Only internally for bookkeeping activities. | 1 | Fast |
| Consortium Blockchain | Some nodes are controlled by pre-selected nodes. | Inter-institutional trade, settlement, or liquidation | $\geq 1$ | Slightly Fast |

### 3.1.4. Drawbacks of Blockchain

As mentioned before, the essential characteristic of blockchain, distribution, can not only verify all transaction information of participating subjects, effectively guaranteeing information authenticity and traceability [36], but also permit each node or user in the blockchain to enjoy the same equal and independent rights to supervise each other. Moreover, due to the Byzantine fault tolerance mechanism, the blockchain can function in an orderly fashion even when the system receives attacks. Thus, there are many well-known domestic and international projects based on blockchain, such as Bitcoin [37] and Ethereum [38], which rely on hardware arithmetic to reach consensus and have the advantage of high security. Although the application of blockchain is booming, it is undeniable that blockchain technology suffers from consensus mechanism security issues, block capacity, efficiency problems, and high hardware cost expenditure. To address the drawbacks of blockchain, this paper focuses on the block efficiency issue, as subsequently shown in Section 3.2.

*3.2. Proposed E-Bidding System*

3.2.1. System Structure of the Blockchain-Based Electronic Bidding System

The structure of our blockchain-based E-bidding system is composed mainly of three layers: the blockchain foundation layer, interface layer, and application layer, as shown in Figure 1 and Table 3. In addition, the big data system (BDS) is applied to assist the blockchain-based E-bidding system in providing reliable decision-making support for all types of transaction issues and mining the bidding fraud evidence.

(1) Blockchain foundation layer: To ensure the reliable operation of upper-layer bidding services, the blockchain foundation layer provides credible infrastructure for upper-layer architecture. Specifically, blockchain automatically executes the pre-defined smart contracts and triggers corresponding algorithms. Meanwhile, it implements the basic functions of data security sharing, such as on-chain data encryption, integrity assurance, and being untamperable.

(2) Interface layer: The interface layer plays a connecting role and provides an interface between the application layer and the blockchain layer, supporting JAVA-Software Development Kit (SDK), GO-SDK, etc. The SDK provides the blockchain address, private key generation, data signature, data uploading, data encryption, smart contract invocation, etc., and the data signature can support both the international and domestic cryptography standards.

(3) Application layer: The application layer is the gate to receive data and handles the business logic of bidding.

(4) Big data systems: BDS is employed to optimize the bidding process for vulnerabilities and avoid bidding fraud. Further, BDS collects data from all stages and can assist the decision-making for all types of transaction issues, while also boosting operational effectiveness and ensuring project quality.

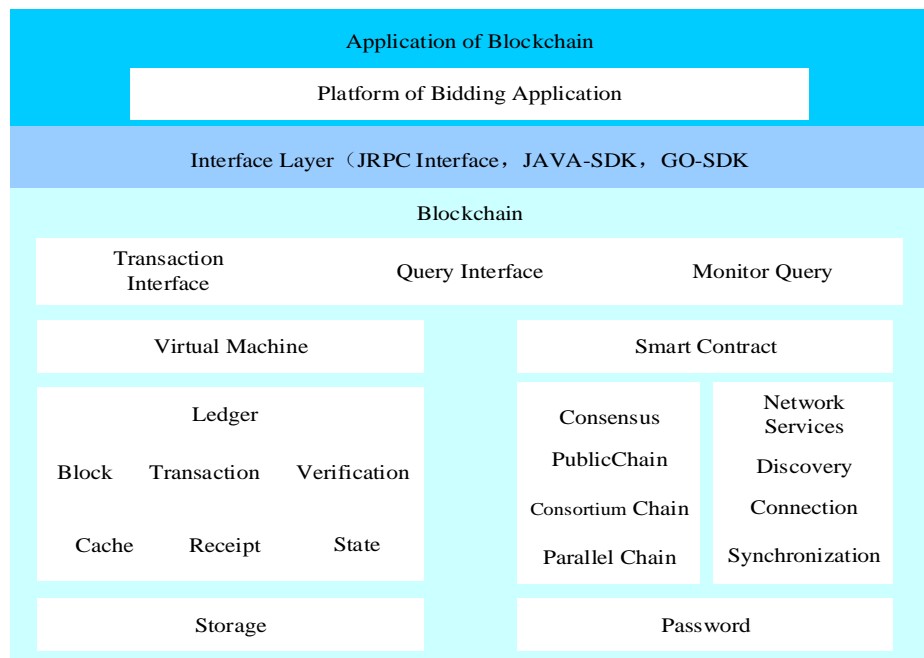

**Figure 1.** Architecture of the blockchain-based E-bidding system.

The main difference between the proposed system and the previous approach is whether blockchain technology and BDS are used. Therefore, we introduce only the blockchain foundation layer and BDS in detail in the following sections.

**Table 3.** The components of blockchain.

| Components | Description |
|---|---|
| Blockchain foundation layer | Ensure the reliable operation of upper-layer bidding services. |
| Interface layer | Create a connection between the blockchain foundation layer and the application layer. |
| Application layer | The gate to receive data and handle the business logic of bidding. |
| Big data systems | Assist blockchain electronic bidding system to optimize the bidding process. |

3.2.2. Structure of the Blockchain Foundation Layer

Blockchain records every key information in each segment, i.e., tenderer information, bid documents, evaluator information, the opening, evaluation, bidding determination, and contract signing. Various data need to be stored, including text, images, and documents, among which text information can be directly stored on the blockchain, while images and documents are usually stored with a hash value that easily suffers from being tampered from attackers. To address this issue, a distributed blockchain node system is the key component to ensure data security. The corresponding hash value will be changed if the original data on the chain is tampered with, which will lead to a data mismatch. This approach can not only solve the cost and efficiency problems of big data storage but also keep the data unchanged.

The blockchain node system consists of consensus nodes, supervisory nodes, and verification nodes, as shown in Figure 2 and Table 4. Specifically, the consensus node is involved in the consensus of the blocks in the business process, which is responsible for the security of the data; the supervisory nodes can conduct statistics on transaction behaviors, identify the true identity of users on the chain, review transactions, and when needed, the supervisory nodes can restrict transactions and freeze accounts by utilizing smart contracts of account management. Verification nodes, which are captured or released at any time, provide network resources as well as verify the validity of blocks, but they cannot become authentication nodes or super nodes.

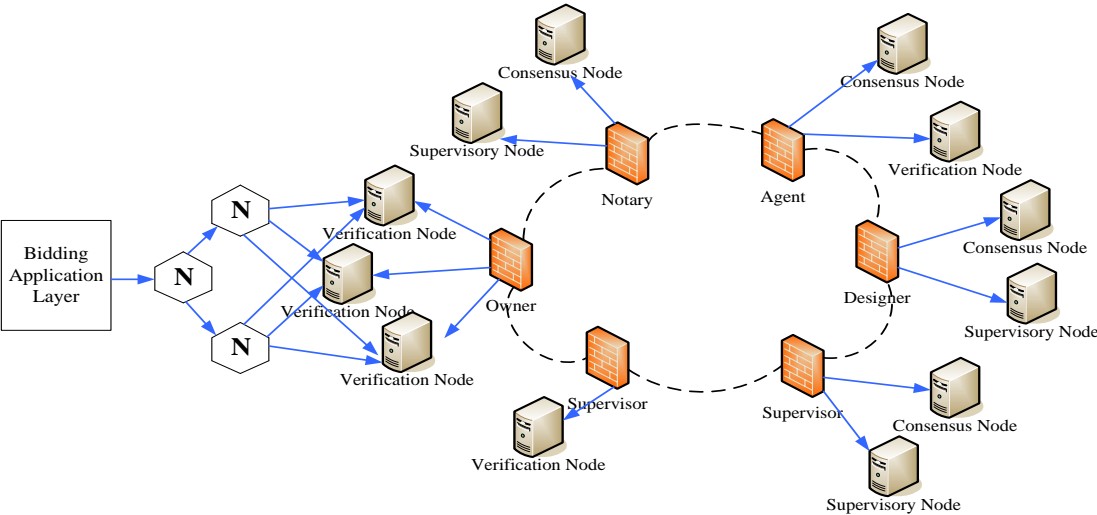

**Figure 2.** The relationship between nodes on the blockchain system.

**Table 4.** The description of nodes.

| Types of Node | Description |
|---|---|
| Consensus nodes | Responsible for the security of the data. |
| Supervisory nodes | Supervise the process of transactions. |
| Verification nodes | Provide network resources as well as verify the validity of blocks. |

The blockchain-based E-bidding system is designed as a consortium chain, which ensures that the traceable data on the chain cannot be tampered with. Moreover, the consensus mechanism of the consortium chain can tolerate node error rates up to one-third, which includes arbitrary node offline and malicious behaviors. Under this mechanism, each node executes the message that it has received most frequently to assure that the node reaches a consistent result; this algorithm is usually called the Byzantine fault tolerance mechanism [39] and is given in Algorithm 1. On the basis of this consortium chain, the consensus mechanism is divided into following parts: proposal phase, pre-selection phase, pre-submission phase, pre-submission waiting phase, submission phase, and block generation phase as shown in Figure 3.

---

**Algorithm 1** Commit

---

**Input:** commitMsg
**Output:** ReplyMsg
1:   **if** verifiedMsg(commitMsg) ! = true
2:     **return** error;
3:     **end procedure**
4:   save commitMsg
5:   **if** state prepared:
6:     **return** ReplyMsg;
7:     **end procedure**
8: **return** none
9: **end procedure**

---

(1)   Proposal phase. The proposal node takes the transaction information out from the Mempool, packs it, and sends the proposal to other validation nodes. Then, the process enters in pre-selection phase.

(2)   Pre-selection phase. Each validation node verifies whether the proposal is legitimate, such as whether the signature is authentic, whether the height is correct, etc. If the proposal passes the verification, it will be transmitted to a pre-selected state.

(3)   Pre-submission phase. If each validation node receives pre-selected messages from more than 2/3 of the other nodes, the process moves on the pre-submission waiting phase.

(4)   Pre-submission waiting phase. If each validation node receives pre-submission messages from more than 2/3 of the other nodes, the process goes to the submission phase.

(5)   Submission phase. The consensus module sends the block to the smart contracts module, which is always regarded as an executor, for a specific execution. Then, when the execution succeeds, the block is stored in the blockchain and ingresses the next phase. After the contract signatory, transaction information is sent to the node's transaction Mempool module through the Remote Procedure Call (RPC) module while it is broadcasted to other nodes through the P2P module to ensure that the transactions of all nodes in the Mempool are consistent at the same time. In summary, the consensus module regularly pulls a list of transaction information from the Mempool, constructs blocks, performs a consensus mechanism, and sends blocks to the executor module to conduct the transactions, which is shown in Figure 4.

(6)   Block generation phase. After the execution, the consensus module sends and writes the block to the Blockchain module. Then, the ledger broadcasts the block to other nodes through the P2P module. After receiving the block, nodes will verify and implement the transactions in the block again and store the block. This phase is shown in Figure 5.

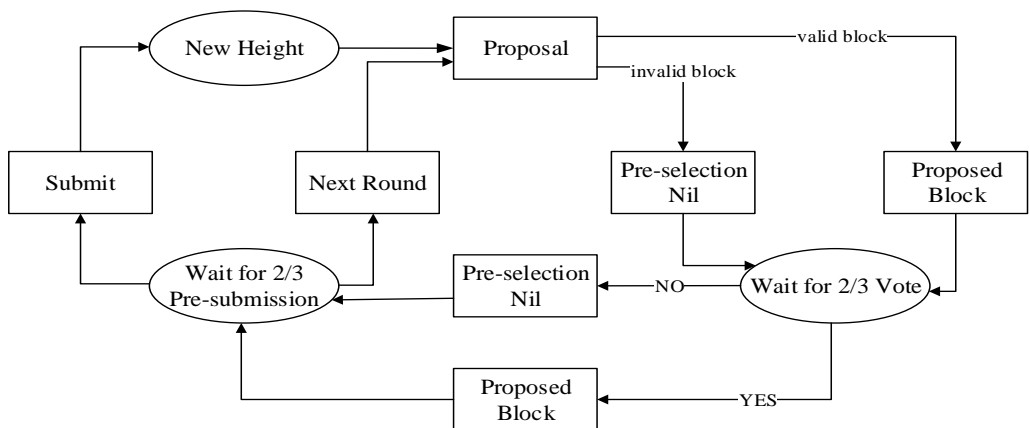

**Figure 3.** Consensus mechanism of consortium chain based on blockchain.

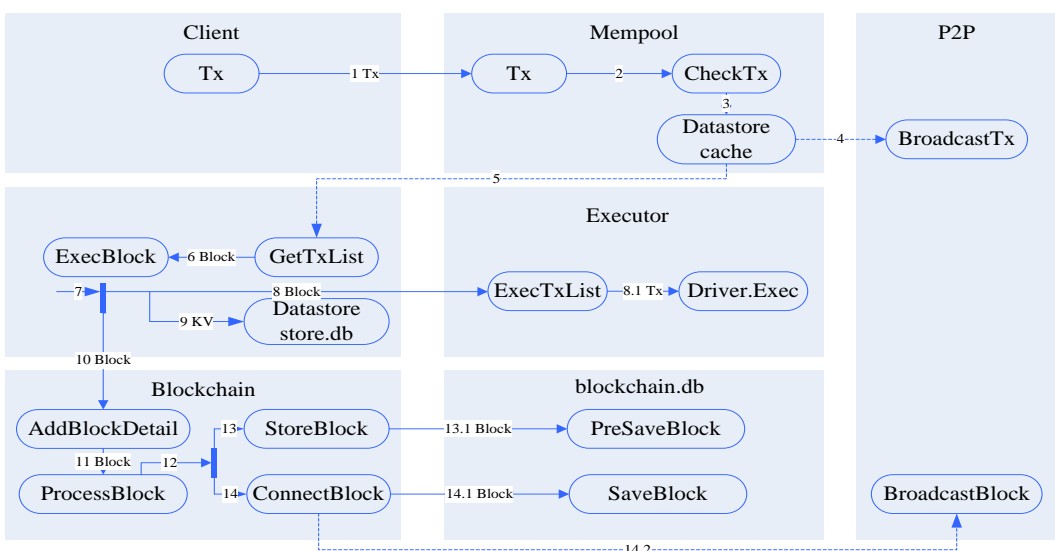

**Figure 4.** Execution of the smart contract module based on blockchain.

### 3.2.3. Process of the Big Data System

Big data technology analyzes the intrinsic linkage of information to quickly locate vulnerabilities in the E-bidding system. Therefore, BDS was designed to further improve the E-bidding system. More prosaically, BDS is organized into two parts: data collection and data analysis, and they will be described in detail in the following section.

Data collection: The main purpose of data collection is to extract valuable data from the entire bidding process, which provides the basis for subsequent analysis. There are three main types of data objects to be collected: data of the tender subject, data generated by the tender process, and evaluation information. Specifically, the data of the tender subject mainly include all types of information including enterprise information and tender information. These data allow a critical quality assessment of companies to limit the number of bidding participants and save running costs. Then, the data generated by the tender process become the main body of data analysis, including information about bid prices, anticipated prices, and expert evaluations, all of which are the most diverse, valuable, and largest part of the data collection phase. Moreover, evaluation information contains mainly contract evaluation and settlement audit information, which is used to supervise the legitimacy of bidding information.

Data analysis: Due to the complexity of large-scale data, it is a significant challenge to process and analyze these data. Thus, an association rule mining algorithm is applied to achieve efficient data analysis. In this stage, we use the frequent itemset mining method

to detect the frequent closed itemsets and provide a minimum description of data fraud evidence, the number of which is between the maximum frequent itemsets and frequent itemsets. To reduce time complexity, we utilize an improved algorithm that mines the maximal frequent itemsets based on the FP tree and solves the problem of frequent itemset updating in bidding fraud data mining. Within the process of frequent itemset mining, the negative and positive association rule mining algorithm is executed, which is practical in solving the conflict between fraud evidence. In addition, the binary support number calculation method is applied to the simple logical operation of "yes" or "no" on the judgment operation of bidding fraud evidence so as to improve the execution efficiency of the algorithm. The progress of bidding fraud evidence mining is shown in Figure 6.

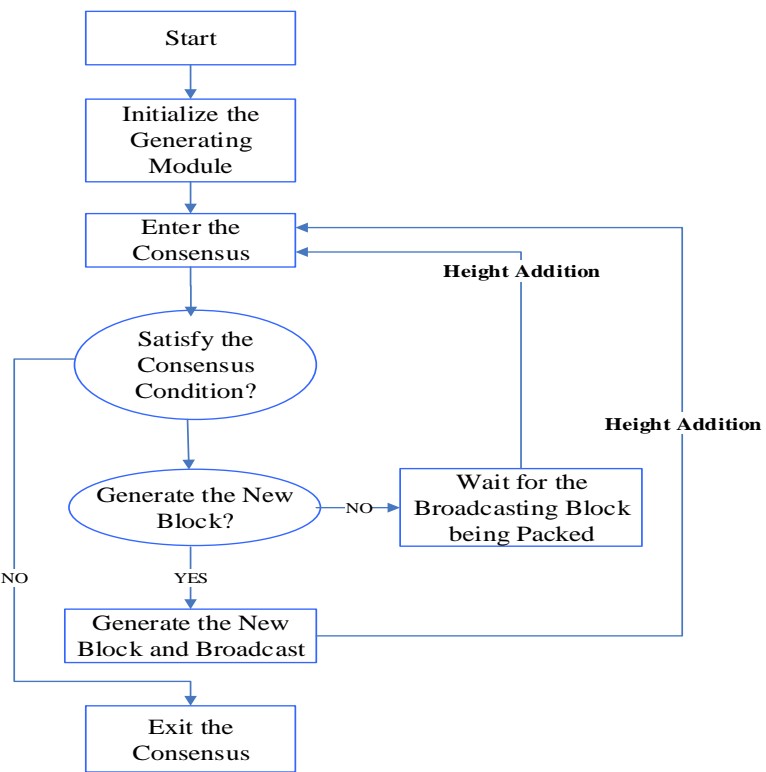

**Figure 5.** Demonstration of the block-generation process by a consensus algorithm.

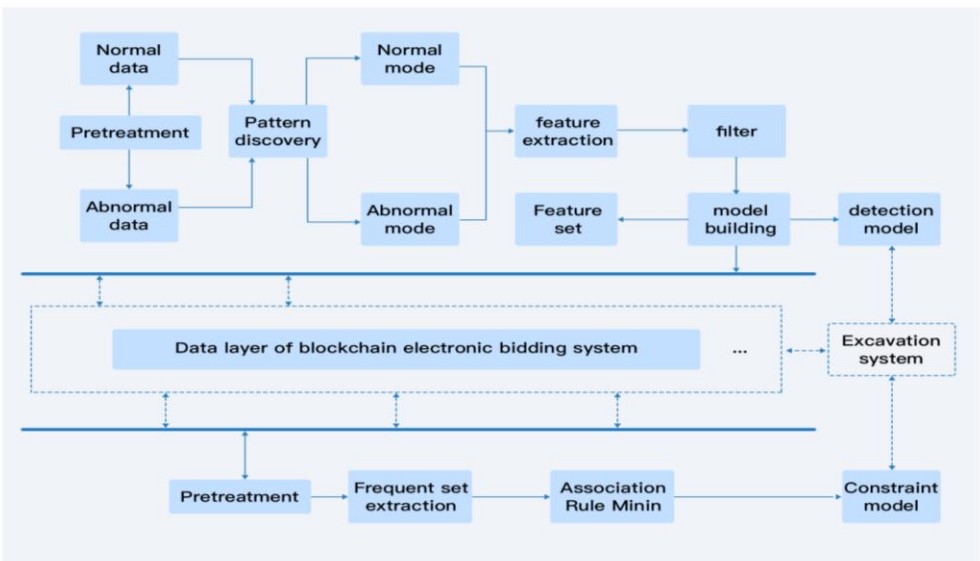

**Figure 6.** Evidence mining framework for bidding fraud.

3.2.4. Overall Processes of the Proposed System

This section describes the process of the blockchain-based E-bidding system in detail, which is divided into six stages: the registration of the tender and tenderer, signing up for the bidding, tender obtaining, tender submitting, tender opening, and the tender deciding and contract management stage, as shown in Figure 7. BDS is embedded into all phases of the E-bidding system and detects bidding fraud data in real-time.

(1) Registration of the Tender and Tenderer. On the blockchain, the corresponding account is assigned to the tenderer. Meanwhile, once uploaded to the chain, tenderers' basic information, such as credentials, credit, and performance, can be permanently stored and cannot be tampered with, and identity information is protected thanks to the blockchain's consensus mechanism. In addition, registration is an optional phase for the designed system, and the basic information of tenderers can be entered at the stage of potential tenderers if registration is not required.

(2) Signing up for the bidding. In this phase, the tenders post the information of specific bidding activity in the designed E-bidding system and this bid document will be stored in the blockchain. If necessary, the key material is encrypted for security. Moreover, the tender will verify the identity of the potential tenderers through blockchain and confirm the results. Each participant in the chain can get specified and reliable bid documents as credentials by using timestamps and produced hash values.

(3) Tender Obtaining Stage. Though the bid document is confidential, the potential tenderer can download and browse these documents to get more bidding details if they pay the bid document fee. Moreover, various previous successful cases are provided to these paid subscribers by the tender authority in the blockchain. Provided tenderers wish to join this bidding activity, they could download the specified bid documents and fill them in online or offline.

(4) Tender Submitting Stage. According to bid requirements and project characteristics, after tenderers complete the bid document, these bid documents will be uploaded to the E-bidding system before the deadline, and the system will automatically anchor the time-point and store the certificate. Due to the high volume of bid documents, a small amount of key information can be encrypted on the chain with specific digital signatures, and the large documents are hashed on the chain, while documents themselves are stored on the file server; this effectively avoids tampering and leakage of important information at the later stage, eliminates irregularities such as tenderer collusion, and ensures a fair and transparent bidding environment.

(5) Tender Opening Stage. Bid evaluators on the blockchain E-bidding system are given corresponding accounts and rights, and their personal data are made available to the public. The P2P and anonymity functions of blockchain can be used to implement P2P transactions, which ensure that remote evaluation of bids can do so impartially and without collusion or favoritism. Within a predetermined amount of time, after authenticating experts' identities on the chain using face or fingerprint recognition, their evaluation results according to the bid document will be stored on the chain.

(6) Tender Deciding and Contract Management Stage. Following the evaluation, the system authorizes the public key of the winning information based on the evaluation results and notifies the winner and the tender to sign the contract online. At the same time, the contract serial number, contract conditions, third-party certification of contract terms, contract subject, and contract filing are all written into the blockchain as witnesses. During the contract public period, any party or supervisory department with concerns about the bidding process can trace the original deposited data of the whole bidding process.

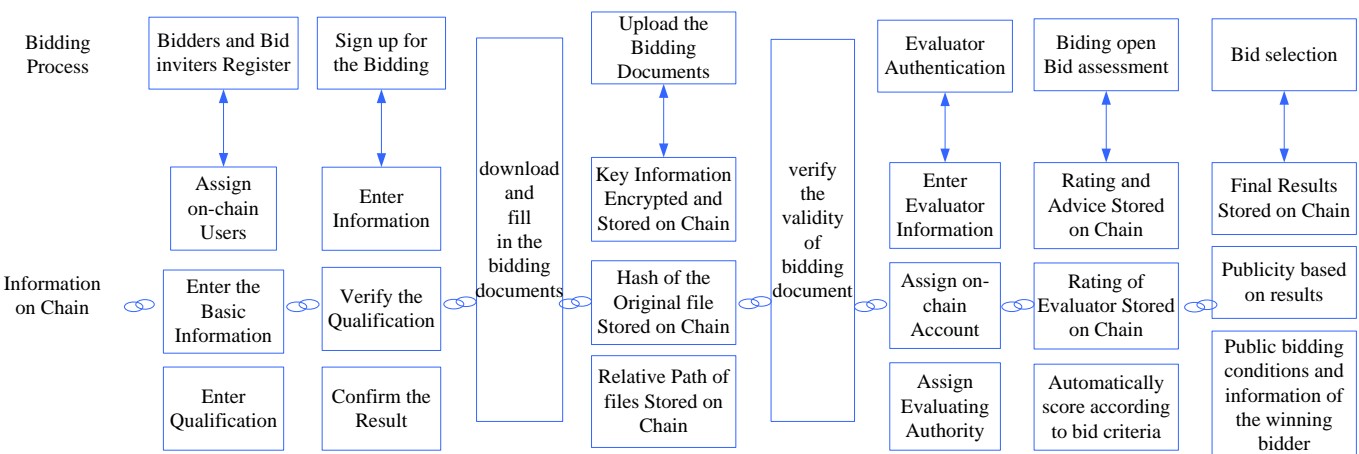

**Figure 7.** The process of blockchain-based E-bidding service.

## 4. Experiments

Our work has developed a decentralized electronic bidding framework based on blockchain technology and big data system and maintained the balance of algorithm complexity and performance to achieve transaction security and privacy protection. By handling bidding data in the big data system, an evidence mining framework for bidding fraud detection is designed, which is applied in the bidding system and has long-term significance for maintaining the fairness of the bidding environment. In this section, we first introduce the experimental settings and platform. We then conduct ablation experiments on the blockchain part to quantitatively evaluate the performance, subsequently compare two encryption algorithms in the proposed framework by designing quantitative and qualitative experiments to analyze the efficiency, test the computation cost of the proposed system, and finally, compare it with other blockchain-based E-bidding systems which are applied in different sectors.

### 4.1. Experimental Settings

We utilize four services as well as a CPU of Intel(R) Xeon(R) Platinum 8378 A and a RAM of 8 G to build the E-bidding system. The system is running on a 64-bit CentOS of version 7.9. As an open-source distributed ledger technology platform, Fabric not only has better performance in transaction processing and transaction confirmation delay but also realizes functions such as smart contracts and confidential transactions. Fabric is an open-source distributed ledger technology platform, and compared with the traditional public chain, it has better performance. Its most important feature is pluggability, and it can be configured to meet as diverse needs as possible. The underlying layer of Fabric consists of peers and orderer nodes that form a P2P network that interacts through Google's open-source RPC framework, gRPC. The middle is isolated using channel technology and each channel is an independent network with its own ledger. Fabric provides gRPC, API, and SDK for upper-layer applications, through which applications can access a variety of resources such as ledger, processing transactions, managing chain-code, registering events, and managing permissions [40]. Therefore, we conduct the experiments with Go language on Fabric, and the run-time calculations are obtained by using the computer system clock.

### 4.2. Ablation Experiment

Generally speaking, the metric of transactions per second is usually used to evaluate the performance of the blockchain. Thus, to validate the performance of the proposed method, we conduct an ablation experiment in terms of transactions per second. In five distinct sets of testing, the average throughput for the proposed system and the system without blockchain are compared in Figure 8. In addition, specific data are displayed in Table 5. Thanks to the parallel mechanism of blockchain, which allows the E-bidding system

to implement several bidding activities at the same time, the transaction throughput of the proposed blockchain-based E-bidding system rises linearly with the number of transactions until it meets the peak at roughly 45 tps, at which point it starts to fall. Moreover, Figure 8 also demonstrates that the proposed methodology is much more effective than the system without blockchain. Specifically, the proposed system can process nearly 24 transactions per second while the system without blockchain can process only up to 11 transactions per second. That is, a system with blockchain technology can double the throughput of the original version method. From this point, it is clear how crucial blockchain technology is to transaction speed.

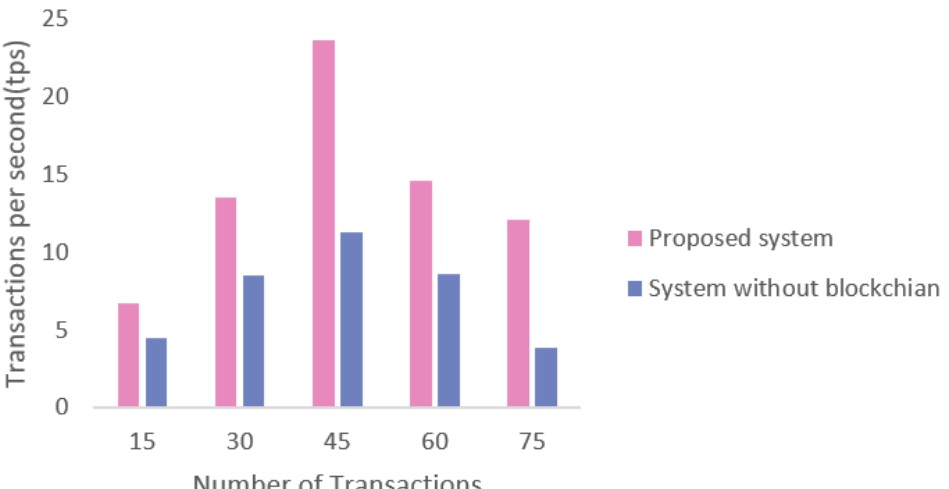

**Figure 8.** Average throughput comparison between proposed system and system without blockchain.

**Table 5.** Average throughput comparison of proposed system with/without blockchain.

| No. of Transactions | Transactions Per Second (tps) | |
|---|---|---|
| | **Proposed System** | **System without Blockchain** |
| 15 | 6.7 | 4.5 |
| 30 | 13.5 | 8.5 |
| 45 | 23.7 | 11.3 |
| 60 | 14.9 | 8.6 |
| 75 | 12.1 | 3.9 |

*4.3. Performance Comparison of Encryption Algorithms*

To a great extent, the efficiency of blockchain depends on the encryption algorithm [41]. Thus, in the proposed framework, we compared the two well-known encryption algorithms, elliptic curve cryptography (ECC) [42] and RSA [43], for time complexity and implementation of transaction validation. The relative pseudocode of ECC and RSA is given in Algorithms 2, 3, 4, and 5, respectively.

---

**Algorithm 2** ECC encryption algorithm

---

**Input:** elliptic curve Ep(a, b), base point G, order n, random integer r, private key k, public key K, plaintext m
**Output:** ciphertexts c1 and c2
1: Select k (k < n)
2: Compute K = k ∗ G
3: Select r (r < n)
4: Compute c1 = m + r ∗ K
5: Compute c2 = r ∗ G
6: **return** c1, c2
7: **end procedure**

---

---

**Algorithm 3** ECC decryption algorithm

---

**Input:** elliptic curve Ep(a, b), base point G, order n, random integer r, private key k, public key K, ciphertexts c1 and c2
**Output:** plaintext m
1: Compute M = c1 − k ∗ c2
2: Encode M
3: **return** M
4: **end procedure**

---

---

**Algorithm 4** RSA encryption algorithm

---

**Input:** public key (x, y), plaintext m
**Output:** ciphertext c
1: Compute $c = m^y \bmod x$
2: **return** c
3: **end procedure**

---

---

**Algorithm 5** RSA decryption algorithm

---

**Input:** public key (x, y), private key k, ciphertext c
**Output:** plaintext m
1: Compute $m = c^k \bmod x$
2: **return** m
3: **end procedure**

---

### 4.3.1. Time Complexity

Table 6 and Figure 9 certainly illustrate that ECC surpasses RSA in terms of time complexity. Even though both the corresponding time complexity of ECC and RSA tend to rise with the number of bits, the time complexity of ECC is consistently lower than that of RSA. The fundamental reason for this is that ECC, as opposed to RSA, better satisfies all the characteristics necessary to meet blockchain security requirements.

**Table 6.** Time complexity comparison of ECC and RSA.

| Number | Time Complexity (ms) | |
| --- | --- | --- |
| | ECC | RSA |
| 1 | 3.5 | 13.8 |
| 2 | 3.8 | 15.2 |
| 3 | 4.6 | 17.3 |
| 4 | 5.2 | 18.9 |
| 5 | 5.8 | 19.7 |

### 4.3.2. Key Size, Encryption Time, and Decryption Time

On the basis of the comparison of ECC and RSA key size, encryption time, and decryption time shown in Table 7, we can observe that while ECC requires fewer bits, RSA has a similar level of protection. Concretely speaking, when RSA needs a 16,358-bit key to provide the resembled security level, ECC employs just a 622-bit key. Furthermore, though the encryption time of ECC is slower than the encryption time of RSA, ECC outperforms RSA in terms of efficiency when considering the decryption time as well. These outcomes are mainly because a shorter key leads to much less CPU and memory consumption as well as faster encryption and decryption time.

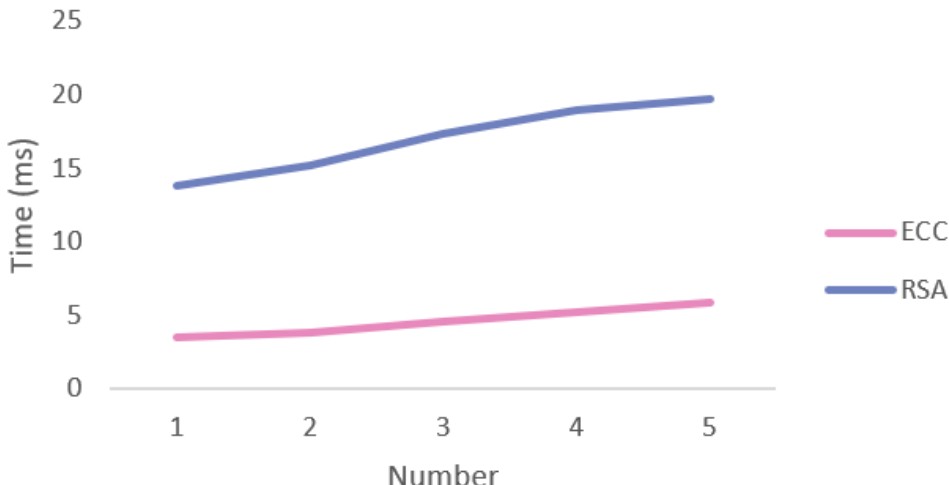

**Figure 9.** Time complexity comparison of algorithms.

**Table 7.** Performance comparison of ECC and RSA.

| Key Size (Byte) | | Encryption Time (s) | | Decryption Time (s) | |
|---|---|---|---|---|---|
| ECC | RSA | ECC | RSA | ECC | RSA |
| 178 | 1223 | 9.59 | 0.69 | 25.01 | 27.62 |
| 251 | 2362 | 61.23 | 0.82 | 25.98 | 121.38 |
| 297 | 3521 | 73.36 | 0.95 | 26.65 | 230.36 |
| 399 | 8353 | 100.26 | 1.24 | 35.01 | 313.67 |
| 622 | 16,358 | 121.35 | 1.62 | 47.91 | 455.61 |

To show the above trend more vividly, we illustrate the data of Table 7 in Figure 10, which also shows that the differences between ECC and RSA are more apparent as the key size grows and under the same degree of protection, RSA needs much more key size than ECC. As can be seen from Table 7, a robust ECC cryptosystem needs keys with a minimum key size of 178 bits. Therefore, we chose key sizes of 178 bits for ECC and 1223 bits for RSA as starting points in Figure 10. Afterward, Figure 10 presents the predominance of ECC.

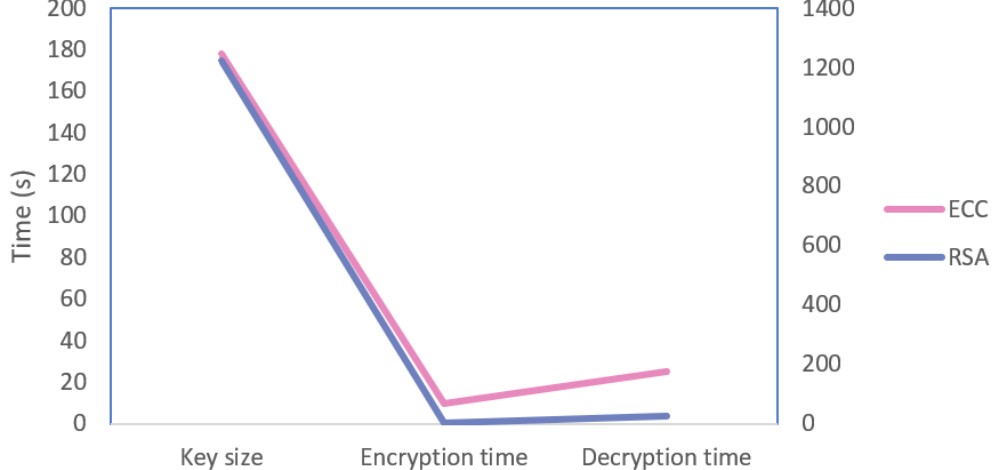

**Figure 10.** Comparison of ECC and RSA in terms of key size, encryption time, and decryption time.

### 4.4. Computation Cost

Additionally, six cases for various tenders with numerous amounts of bids are tested. Table 8 and Figure 11 reveal that even with 41 tenders and 70 bids, the computation cost is only 72.353 ms, which indicates that the adoption of big data technology can substantially

decrease the large resource consumption and enhance the effectiveness of the proposed system. As a result, high performance can be attained by implementing our framework.

**Table 8.** Computation cost of the blockchain.

| No. of Case | Tenders | Bids | Computation Cost (ms) |
|---|---|---|---|
| Case 1 | 9 | 10.5 | 12.12 |
| Case 2 | 13.65 | 20.7 | 25.27 |
| Case 3 | 19.36 | 34.98 | 39.79 |
| Case 4 | 25.78 | 49.71 | 56.25 |
| Case 5 | 36.352 | 65.57 | 62.291 |
| Case 6 | 41.695 | 70.39 | 72.353 |

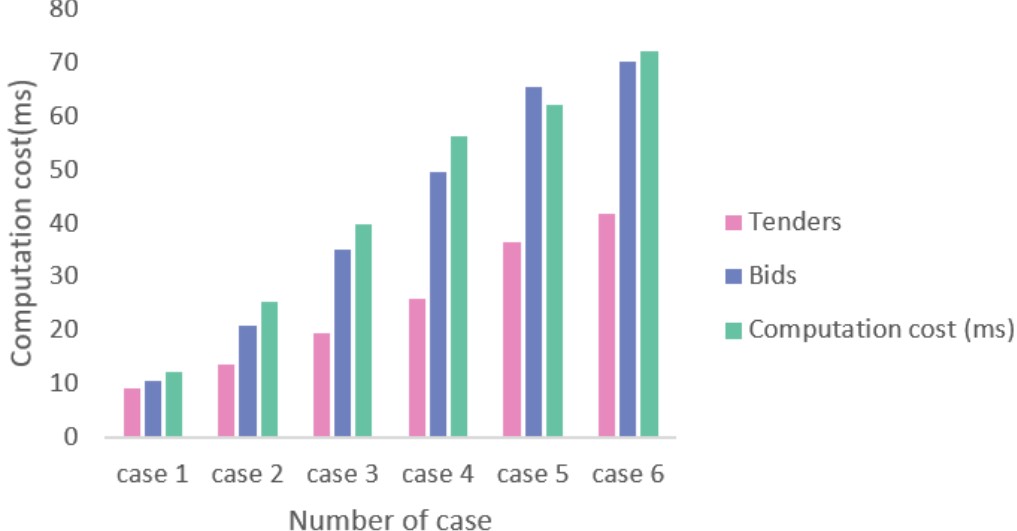

**Figure 11.** Computation cost with various tenders and bids.

### 4.5. Comparisons of Security with The-State-of-the-Art Methods

To evaluate the performance of the proposed bidding system, four popular systems are utilized, namely, those of Chen [44], Nair [45], Johnson [46], and Wang [47]. Additionally, six metrics that are necessary for an E-voting system are adopted to exhibit comprehensive comparisons, i.e., completeness, anonymity, fairness, eligibility, rationality, and non-repeatability. Specifically, the meanings of these metrics are provided as follows.

Completeness: Completeness is when each person can check whether the bidding information is correct.

Anonymity: Anonymity ensures that no internal or external attackers can know the identity and transactions of other people.

Fairness: A technology or protocol that does not discriminate against the honest and correctly participating members is said to be fair.

Eligibility: Eligibility means that only those with legal qualifications have access to the system to protect the fairness of the voting or bidding process.

Rationality: Rationality denotes that no internal or external attackers have the opportunity to maliciously tamper with other people's bidding, thereby ensuring the legitimacy of the voting process.

Non-repeatability: Non-repeatability denotes that each operation is done only once.

As can be seen in Table 9, our system has more comprehensive security than several systems, which is based on the following merits. (1) In our system, every node verifies whether the new data is correct through the existing data on the blockchain. Due to the great difficulty in tampering with existing information and the closeness of the blockchain system, completeness is ensured. (2) Our system ensures the traceability and anonymity of

data through an encryption algorithm. (3) We use a decentralized consensus mechanism that makes each member's encrypted identity and bidding information public to other members for verification, which can also reflect fairness, to a great extent. (4) Our big data system could filter out malicious bidding attacks. Furthermore, in the registration stage and bidding information transferring procedures, our system verifies and encrypts the identity information of tender and tenderers to guarantee eligibility. (5) In our scheme, if individuals want to tamper with the information of a block on the blockchain, they must lead a new branch from the block and create a new chain that exceeds the length of the original chain, which is computationally impossible. (6) In our system, the blockchain prevents double-bidding by timestamping groups of transactions and then broadcasting them to all of the nodes in the system. As operations are time-stamped on the blockchain and mathematically related to the previous ones, they are irreversible and impossible to tamper with.

**Table 9.** Comparison of security properties.

| Method | Completeness | Anonymity | Fairness | Eligibility | Rationality | Non-Repeatability |
|---|---|---|---|---|---|---|
| Chen's [44] | ✓ | ✓ | ✗ | ✓ | ✓ | ✓ |
| Nair's [45] | ✗ | ✗ | ✓ | ✗ | ✓ | ✗ |
| Johnson's [46] | ✗ | ✓ | ✓ | ✓ | ✓ | ✓ |
| Wang's [47] | ✓ | ✓ | ✓ | ✓ | ✓ | ✗ |
| Ours | ✓ | ✓ | ✓ | ✓ | ✓ | ✓ |

In summary, our proposed bidding system is very beneficial for improving security, data traceability, and cooperation.

## 5. Conclusions

This paper proposes an implementation path and method of blockchain technology to solve the existing problems of electronic bidding system, which provides a realistic solution for solving the design standardization of electronic bidding platforms, system security and stability, and traceability and storage of bidding process. In addition, through this paper, practitioners related to electronic bidding can understand the latest research trends and technological innovation methods of blockchain technology in this field and become familiar with the main problems and technical paths solved by blockchain technology in electronic bidding. At present, there is still a gap in research in this field at home and abroad, and this paper is of great significance for blockchain technology to empower the industrialization, industrialization, and digitalization of construction, and promote the transformation and upgrading of the construction industry.

**Author Contributions:** Conceptualization, D.X. and Q.Y.; methodology, D.X.; software, D.X.; validation, D.X. and Q.Y.; formal analysis, D.X.; investigation, D.X.; data curation, D.X.; writing—original draft preparation, D.X.; writing—review and editing, Q.Y.; visualization, Q.Y.; supervision, D.X.; project administration, D.X.; funding acquisition, D.X. All authors have read and agreed to the published version of the manuscript.

**Funding:** This research was funded by the 2020 Science and Technology Plan Project of the Ministry of Housing and Urban-Rural Development, under Project Number 2020-K-061.

**Data Availability Statement:** Not applicable.

**Acknowledgments:** Gao Baojian of Jiangsu Construction Engineering Group Co., Ltd. who also contributed to this article. We would like to express our gratitude to Baojian Gao.

**Conflicts of Interest:** The authors declare no conflict of interest.

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
