# Peer review of "The Systems Approach and Design Path of Electronic Bidding Systems Based on Blockchain Technology"

_electronics, doi:10.3390/electronics11213501_

Round 1
Reviewer 1 Report (Previous Reviewer 4)
This paper proposes a blockchain-based e-bidding framework and a big data system to improve the efficiency and security of the bidding process in the construction industry. It collects large amounts of data and analyzes it in order to determine whether there is bidding fraud or collusion and to provide a fair digital "evidence". How do you define fair? I intentionally use that word from the abstract here. Why evidence is in “”? You do have digital proofs.
The paper needs a proof-reading.
line 30 - update the sentence for COVID
line 77 - write contribution in plural
Table 5 - replace [ms] with [s], the unit does not correspond to the written text
Figure 2 is not precise enough. Either delete it or replace with a more comprehensible one [1]. Add state-of-the-art that explains well the text below figure 2 [1]. At the moment is not written precisely.
[1] M. Raikwar, et al., "SoK of Used Cryptography in Blockchain," in IEEE Access, vol. 7, pp. 148550-148575, 2019, doi: 10.1109/ACCESS.2019.2946983.
How do you get a consent for collecting the data?
The authors include quantitative and qualitative experiments to demonstrate the performance of the proposed system.
Author Response
Response to Reviewer 1 Comments
This paper proposes a blockchain-based e-bidding framework and a big data system to improve the efficiency and security of the bidding process in the construction industry. It collects large amounts of data and analyzes it in order to determine whether there is bidding fraud or collusion and to provide a fair digital "evidence". How do you define fair? I intentionally use that word from the abstract here. Why evidence is in “”? You do have digital proofs.
Response1:whether there is bidding fraud or collusion or not and provide valid digital "evidence", which contributes greatly to the fairness and transparency of the bidding process.
The paper needs a proof-reading.
line 30 - update the sentence for COVID
Response2:Besides, since the emergence of COVID-19 to the spread of Omikron in the context of the global scale, it is of great practical significance to study how to realize the whole process of E-bidding online and how to ensure continuous economic activities and reduce personnel contact.
line 77 - write contribution in plural
Response3:We change the sentence: The main contributions of this work can be summarized as follows.
Table 5 - replace [ms] with [s], the unit does not correspond to the written text
Response4:It has been confirmed and should be ms.
Figure 2 is not precise enough. Either delete it or replace with a more comprehensible one [1]. Add state-of-the-art that explains well the text below figure 2 [1]. At the moment is not written precisely.
- Raikwar, et al., "SoK of Used Cryptography in Blockchain," in IEEE Access, vol. 7, pp. 148550-148575, 2019, doi: 10.1109/ACCESS.2019.2946983.
Response5:Already delete.
How do you get a consent for collecting the data?
Response6:The data used in this paper are all based on the main data formed in the research and development process of the Ministry of Housing and Urban-Rural Development, which is mainly completed through enterprise behavior. The project leader is the first author of this paper. Therefore, the data collected in this paper are all from independent intellectual property rights, and there is no property rights dispute.
The authors include quantitative and qualitative experiments to demonstrate the performance of the proposed system.
Response7:See previous reply, please.

Reviewer 2 Report (New Reviewer)
The paper is interesting.
The related work shall be strengthened. Does the paper describe and evaluate previous literature on the subject? Do the authors show awareness of work that has been published recently in the area?
Authors are suggested to review more new and relevant research to support their research contribution. The following references are recommended for possible consideration:
Blockchain technology
Ferrández-Pastor, F. J., Mora-Pascual, J., & Díaz-Lajara, D. (2022). Agricultural traceability model based on IoT and Blockchain: Application in industrial hemp production. Journal of Industrial Information Integration, 29, 100381.
Munim, Z. H., Balasubramaniyan, S., Kouhizadeh, M., & Hossain, N. U. I. (2022). Assessing blockchain technology adoption in the Norwegian oil and gas industry using Bayesian Best Worst Method. Journal of Industrial Information Integration, 28, 100346.
Gorkhali, A., & Chowdhury, R. (2022). Blockchain and the evolving financial market: A literature review. Journal of Industrial Integration and Management, 7(01), 47-81.
Big data
Duan, L., & Xiong, Y. (2015). Big data analytics and business analytics. Journal of Management Analytics, 2(1), 1-21.
Hassani, H., Huang, X., & Silva, E. (2018). Banking with blockchain-ed big data. Journal of Management Analytics, 5(4), 256-275.
Javaid, M., Haleem, A., Singh, R. P., & Suman, R. (2021). Significant applications of big data in Industry 4.0. Journal of Industrial Integration and Management, 6(04), 429-447.
Author Response
Response to Reviewer 2 Comments
Comments and Suggestions for Authors
The paper is interesting.
The related work shall be strengthened. Does the paper describe and evaluate previous literature on the subject? Do the authors show awareness of work that has been published recently in the area?
Authors are suggested to review more new and relevant research to support their research contribution. The following references are recommended for possible consideration:
Blockchain technology
Ferrández-Pastor, F. J., Mora-Pascual, J., & Díaz-Lajara, D. (2022). Agricultural traceability model based on IoT and Blockchain: Application in industrial hemp production. Journal of Industrial Information Integration, 29, 100381.
Munim, Z. H., Balasubramaniyan, S., Kouhizadeh, M., & Hossain, N. U. I. (2022). Assessing blockchain technology adoption in the Norwegian oil and gas industry using Bayesian Best Worst Method. Journal of Industrial Information Integration, 28, 100346.
Gorkhali, A., & Chowdhury, R. (2022). Blockchain and the evolving financial market: A literature review. Journal of Industrial Integration and Management, 7(01), 47-81.
Big data
Duan, L., & Xiong, Y. (2015). Big data analytics and business analytics. Journal of Management Analytics, 2(1), 1-21.
Hassani, H., Huang, X., & Silva, E. (2018). Banking with blockchain-ed big data. Journal of Management Analytics, 5(4), 256-275.
Javaid, M., Haleem, A., Singh, R. P., & Suman, R. (2021). Significant applications of big data in Industry 4.0. Journal of Industrial Integration and Management, 6(04), 429-447.
Response1:We thoroughly studied the references given by the experts and cited them in the text. The reference is 3 4 15 16 17.

Reviewer 3 Report (New Reviewer)
The authors proposed an E-bidding system with blockchain technology. The overall system design is shown in Figure 3 on page 7 and Figure 9 on page 13. There are some comments I hope can help the papers become better:
- The authors wrote about big data in the introduction of the article, but I can't find any big data experimental in the experiment section. I think the system just only a general data system with blockchain technology, so maybe just don't add Big Data System as a feature.
- In Figure 10 and Table 5, the authors wrote with the y-axis title "Transactions per second (ms)" the transactions are in ms unit. Please double confirm.
- In subsection 4.1, "experimental settings," the authors mentioned four services. Please note the specific services that the authors apply.
- Moreover, maybe the authors need to reference which fabric that authors apply in the system. A brief introduction is required.
- In Algorithm 3, the return value m doesn't exist. Is it uppercase M?
- Lack of unit and title in some figures and tables, for example, Table 7's Key Size; Figure 13's x and y axis title.
- The conclusion and abstract are too weak and didn't mention some critical findings. The authors do many experiments on the algorithm's performance and cost or blockchain but didn't summarize the result in conclusion.
- Please check that the Table and Figure arrangement should show after reference in the paragraph; for example, Table 2 should arrange to the end of 3.1.3.
- The English writing needs to improve; for example, in line 36, "Secondly, affected by network security...." the subject is gone.
Author Response
Response to Reviewer 3 Comments
Comments and Suggestions for Authors
The authors proposed an E-bidding system with blockchain technology. The overall system design is shown in Figure 3 on page 7 and Figure 9 on page 13. There are some comments I hope can help the papers become better:
- The authors wrote about big data in the introduction of the article, but I can't find any big data experimental in the experiment section. I think the system just only a general data system with blockchain technology, so maybe just don't add Big Data System as a feature.
Response1:This paper points to an applied research, in the process of designing and developing an electronic bidding system of blockchain technology, involving the statistics, analysis and feedback of registered users, including: government functional management departments, industry management departments, construction units, bidding agencies, potential bidding units, notary institutions and related potential participants. The design of the big data model is intended to reflect the changes of the actors in the blockchain technology architecture system in the block nodes of any bidding project, and at the same time reflect the behavior of the bidder and the bidder (including registration information, transmission of text information, modification of text information, server address change, network authentication, etc.) data model situation. However, this data model has not yet been empirically applied, so the content of big data technology will not be highlighted in some parts of keywords and articles.
- In Figure 10 and Table 5, the authors wrote with the y-axis title "Transactions per second (ms)" the transactions are in ms unit. Please double confirm.
Response2:checking have been made.
- In subsection 4.1, "experimental settings," the authors mentioned four services. Please note the specific services that the authors apply.
Response3:Modifications have been made.
- Moreover, maybe the authors need to reference which fabric that authors apply in the system. A brief introduction is required.
Response4:The question is relatively general, and we are not sure whether the experts are asking about the system structure of electronic bidding or the system structure of the underlying architecture of the blockchain. The structure of the blockchain is illustrated in Figure 1.
- In Algorithm 3, the return value m doesn't exist. Is it uppercase M?
Response5:Modifications have been made.
- Lack of unit and title in some figures and tables, for example, Table 7's Key Size; Figure 13's x and y axis title.
Response6:Modifications have been made.
- The conclusion and abstract are too weak and didn't mention some critical findings. The authors do many experiments on the algorithm's performance and cost or blockchain but didn't summarize the result in conclusion.
Response7:See new Abstract, please.
The electronic tendering and bidding system has realized the digitalization, networking and highly integration of the whole process of tendering, bidding, bid evaluation and contract, which has a wide range of applications. However, the trust degree, cooperation and transaction efficiency of the parties involved in electronic bidding are low, and bidding fraud and collusion are forbidden repeatedly. Blockchain technology has the characteristics of decentralization, transparent transactions, traceable, non-tampering and forgery, and data security. This paper proposes a design path of electronic bidding system based on blockchain technology, which aiming to solve the efficiency, trust and security of electronic trading process. By building the underlying architecture platform of blockchain and embedding the business process of electronic bidding, which realizing the transparent, opening and traceable during the whole process of electronic bidding. This paper uses qualitative and quantitative methods to prove the effectiveness of the system.
See the new conclusion
This paper proposes an implementation path and method of blockchain technology to solve the existing problems of electronic bidding system, which provides a realistic solution for solving the design standardization of electronic bidding platform, system security and stability, traceability and storage of bidding process, and practitioners related to electronic bidding can understand the latest research trends and technological innovation methods of blockchain technology in this field through this paper, and be familiar with the main problems and technical paths solved by blockchain technology in electronic bidding. At present, there is still a gap in research in this field at home and abroad, and this paper is of great significance for blockchain technology to empower the industrialization, industrialization and digitalization of construction, and promote the transformation and upgrading of the construction industry.
- Please check that the Table and Figure arrangement should show after reference in the paragraph; for example, Table 2 should arrange to the end of 3.1.3.
Response8:The full text chart has been reformatted according to the comments.
- The English writing needs to improve; for example, in line 36, "Secondly, affected by network security...." the subject is gone.
Response9:Secondly, the E-bidding system is difficult to ensure the identity authentication of users and data security, which affected by network security.

Reviewer 4 Report (New Reviewer)
1. Delete the “research on” from the title.
2. Restructure the abstract. It should contain the issues, objectives, methods, results and implications.
3. I wonder why the construction industry's term needs to have a reference. I suggest the authors carefully place the reference information right after the statement. Most of it was incorrect.
4. There is a need for a logical statement and supporting reference that argues blockchain technology is fit to support the E-bidding system.
5. It is not really clear why combining blockchain technology and an E-bidding system can propose a blockchain-based E-bidding system applied in the construction industry. It should be the E-bidding system should be supported by blockchain technology. However, how could it be done?
6. The research gaps are not supporting the contribution. So please develop this part properly.
7. Section 2.1 should be discussed the functionalities and facilities of the Electronic-Bidding System in the Construction Industry. However, the current review is too general.
8. It is unclear how the authors include Big Data Technology in the model. For example, which one focus on the study? Why not discuss it in the gaps and objectives?
9. Section 3 should use the method instead of the Proposed Methodology.
10. Figure 2 needs further elaboration.
11. Is there any reference for Table 1?
12. Figure 9. The process of blockchain-based E-bidding service needs to include a verification mechanism.
13. What can you conclude from Table 5? Any difference between with and without?
14. Include implications to the practitioners and future studies.
15. Discuss the limitation properly.
Author Response
Response to Reviewer 4 Comments
Comments and Suggestions for Authors
- Delete the “research on” from the title.
Response1:The Systems Approach and Design Path of Electronic Bidding Systems Based on Blockchain Technology
- Restructure the abstract. It should contain the issues, objectives, methods, results and implications.
Response2:See new Abstract, please.
The electronic tendering and bidding system has realized the digitalization, networking and highly integration of the whole process of tendering, bidding, bid evaluation and contract, which has a wide range of applications. However, the trust degree, cooperation and transaction efficiency of the parties involved in electronic bidding are low, and bidding fraud and collusion are forbidden repeatedly. Blockchain technology has the characteristics of decentralization, transparent transactions, traceable, non-tampering and forgery, and data security. This paper proposes a design path of electronic bidding system based on blockchain technology, which aiming to solve the efficiency, trust and security of electronic trading process. By building the underlying architecture platform of blockchain and embedding the business process of electronic bidding, which realizing the transparent, opening and traceable during the whole process of electronic bidding. This paper uses qualitative and quantitative methods to prove the effectiveness of the system.
- I wonder why the construction industry's term needs to have a reference. I suggest the authors carefully place the reference information right after the statement. Most of it was incorrect.
Response3:Already Deleted
- There is a need for a logical statement and supporting reference that argues blockchain technology is fit to support the E-bidding system.
Response4:The electronic bidding and tendering system has the business functions of online bidding, bidding, bid evaluation, contract management and so on, involving the business subjects such as tenderee, bidding agency, potential bidder, notary agency and government regulatory department, realizing the digitalization, networking and integration of the whole process of business. However, under general conditions, electronic bidding system is provided by government agencies, other market participants competition notary and transparency is not enough, lack of mutual trust and coordination, resulting in low transaction efficiency, poor fairness, bidding fraud and collusion behavior is also repeatedly banned. The characteristics of blockchain technology, such as decentralization, transaction transparency, traceable, non-tampering and data security, can make up for the defects of the electronic bidding system, and realize the fairness, justice and transparency of the electronic bidding system in a larger market scope, more participants and more application scenarios.
- It is not really clear why combining blockchain technology and an E-bidding system can propose a blockchain-based E-bidding system applied in the construction industry. It should be the E-bidding system should be supported by blockchain technology. However, how could it be done?
Response5:See previous reply, please.
- The research gaps are not supporting the contribution. So please develop this part properly.
Response6:Our research has found that there is relatively little research literature in this area.
- Section 2.1 should be discussed the functionalities and facilities of the Electronic-Bidding System in the Construction Industry. However, the current review is too general.
Response7:The traditional project bidding field has gone through a long road of development under the norms of laws and regulations such as the "Tendering and Bidding Law" and the "Government Procurement Law", which has played an important role in unifying the rules of the bidding market and encouraging orderly competition in the market. The emergence and wide application of the Internet is a revolution in industrial society, for the construction industry in the field of engineering bidding, the emergence and development of electronic bidding has also redefined the ways and methods of bidding by construction market entities, and has played a positive role in further promoting a free, fair, just and honest market environment.In recent years, the government and relevant industry organizations have supported and encouraged construction units and relevant market entities to carry out electronic bidding and bidding work, which has effectively promoted the application of electronic bidding. The electronic bidding system realizes business functions such as online bidding, bidding, bid evaluation, and contract management, reduces offline transaction costs, improves work efficiency, enhances the information management capabilities of governments and participating entities, and effectively promotes the digitalization, networking and high integration of the whole bidding process.However, there are still some problems with the current electronic tendering, resulting in the application of electronic tendering still being quite limited. First, relevant laws and regulations lag behind, lack of unified norms and standards, and it is difficult to promote; Second, there are many electronic bidding platforms, which are poorly compatible with each other, and the phenomena of administrative intervention and secret operations cannot be effectively prevented; Third, the security and stability of the electronic bidding platform need to be strengthened, if the data security and stability performance is not effectively guaranteed, it is very easy to cause the leakage of commercial secrets and malicious tampering of data information. The research on these problems is of great practical significance for the application of electronic bidding in the construction industry.
- It is unclear how the authors include Big Data Technology in the model. For example, which one focus on the study? Why not discuss it in the gaps and objectives?
Response8:This paper points to an applied research, in the process of designing and developing an electronic bidding system of blockchain technology, involving the statistics, analysis and feedback of registered users, including: government functional management departments, industry management departments, construction units, bidding agencies, potential bidding units, notary institutions and related potential participants. The design of the big data model is intended to reflect the changes of the actors in the blockchain technology architecture system in the block nodes of any bidding project, and at the same time reflect the behavior of the bidder and the bidder (including registration information, transmission of text information, modification of text information, server address change, network authentication, etc.) data model situation. However, this data model has not yet been empirically applied, so the content of big data technology will not be highlighted in some parts of keywords and articles.
- Section 3 should use the method instead of the Proposed Methodology.
Response9:Change Proposed Methodology into Method
- Figure 2 needs further elaboration.
Response10:Already delete.
- Is there any reference for Table 1?
Response11:References have been added.
- Figure 9. The process of blockchain-based E-bidding service needs to include a verification mechanism.
Response12:Already redraw.
- What can you conclude from Table 5? Any difference between with and without?
Response13:When the platform is larger and the more participants, the benefits will be greater and the speed will be faster.
- Include implications to the practitioners and future studies.
Response14:This paper proposes an implementation path and method of blockchain technology to solve the existing problems of electronic bidding system, which provides a realistic solution for solving the design standardization of electronic bidding platform, system security and stability, traceability and storage of bidding process, and practitioners related to electronic bidding can understand the latest research trends and technological innovation methods of blockchain technology in this field through this paper, and be familiar with the main problems and technical paths solved by blockchain technology in electronic bidding. At present, there is still a gap in research in this field at home and abroad, and this paper is of great significance for blockchain technology to empower the industrialization, industrialization and digitalization of construction, and promote the transformation and upgrading of the construction industry.
- Discuss the limitation properly.
Response15:See previous reply, please.

Round 2
Reviewer 3 Report (New Reviewer)
The authors modified the major issues. There are still some minor comments provided below:
- The authors wrote the "fabric" in subsection "4.1. Experimental Settings". Maybe the reference to the open-source distributed ledger "fabric" need to be added and described more to make it more apparent to the reader.
- In Figure 10 and Table 5, the authors wrote with the y-axis title "Transactions per second (ms)" the transactions are in ms unit. I don't think authors should name it transactions per second (TPS). TPS should be the number of transactions per second, not the duration of time. The authors should modify.
- In subsection 4.1, "experimental settings," the authors mentioned four services before. Why deleted? Maybe the authors can indicate more information about experimental settings as well.
- In subsection 4.3.1 Table 6 and Figure 11, please mention the header in the first column. Is it the number of bits?
- In Figure 12, please mention the right-hand side y-axis title and unit.
Author Response
Comments and Suggestions for Authors
The authors modified the major issues. There are still some minor comments provided below:
- The authors wrote the "fabric" in subsection "4.1. Experimental Settings". Maybe the reference to the open-source distributed ledger "fabric" need to be added and described more to make it more apparent to the reader.
Respond 1:We have cited and explained the concept with references. Please see reference:[40]ZHAO Hui-qun, ZHANG Long-long. An architecture evolution algorithm for Fabric blockchain application software. Software.2020, Vol. 41, No. 7;pp.1-10,60.
Fabric is an open source distributed ledger technology platform, compared with the traditional public chain, which has better performance. Its most important feature is pluggability and can be configured to meet as diverse needs as possible. The underlying layer of Fabric consists of peers and orderer nodes that form a P2P network that interacts through Google's open-source RPC framework-gRPC. The middle is isolated using channel technology and each channel is an independent network with its own ledger. Fabric provides gRPC, API and SDK for upper-layer applications, through which applications can access a variety of resources such as ledger, process transactions, manage chain-code, register events, and manage permissions.
- In Figure 10 and Table 5, the authors wrote with the y-axis title "Transactions per second (ms)" the transactions are in ms unit. I don't think authors should name it transactions per second (TPS). TPS should be the number of transactions per second, not the duration of time. The authors should modify.
Respond 2:Change the "Transactions per second (ms)" into “Transactions per second (TPS)”
Table 5. Average throughput comparison of proposed system with/without blockchain.
No. of Transactions |
Transactions per second(tps) |
|||
Proposed system |
System without blockchain |
|||
15 |
6.7 |
4.5 |
||
30 |
13.5 |
8.5 |
||
45 |
23.7 |
11.3 |
||
60 |
14.9 |
8.6 |
||
75 |
12.1 |
3.9 |
Figure 8. Average throughput Comparison between proposed system and system without blockchain.
- In subsection 4.1, "experimental settings," the authors mentioned four services before. Why deleted? Maybe the authors can indicate more information about experimental settings as well.
Respond 3:Considering that the reader may not be clear enough about the "four services" logically, we have deleted it last time. However, considering the continuity and completeness of the article, we will add the content of "four services" this time.The four services are blockchain basic layer, interface layer, application layer and big data system.
- In subsection 4.3.1 Table 6 and Figure 11, please mention the header in the first column. Is it the number of bits?
Respond 4: We change the Table 6 and Figure 11. According to your revision, we change the Figure 9 too, which added the x-axis title.
Table 6. Time complexity comparison of ECC and RSA.
|
Time Complexity (ms) |
|
Number |
ECC |
RSA |
1 |
3.5 |
13.8 |
2 |
3.8 |
15.2 |
3 |
4.6 |
17.3 |
4 |
5.2 |
18.9 |
5 |
5.8 |
19.7 |
Figure 9. Time complexity comparison of algorithms.
Table 8. Computation cost of the blockchain.
No. of case |
Tenders |
Bids |
Computation cost (ms) |
Case 1 |
9 |
10.5 |
12.12 |
Case 2 |
13.65 |
20.7 |
25.27 |
Case 3 |
19.36 |
34.98 |
39.79 |
Case 4 |
25.78 |
49.71 |
56.25 |
Case 5 |
36.352 |
65.57 |
62.291 |
Case 6 |
41.695 |
70.39 |
72.353 |
Figure 11. Computation cost with various tenders and bids
- In Figure 12, please mention the right-hand side y-axis title and unit.
Respond 5: There is no Figure 12. Maybe you means Figure 11? The figure 11 has revised. Please see the respond 4.

Reviewer 4 Report (New Reviewer)
I have carefully read the revised paper submitted by the authors. I think it is suitable for publication in The Electronics.
Author Response
Comments and Suggestions for Authors
I have carefully read the revised paper submitted by the authors. I think it is suitable for publication in The Electronics.
Respond 1: It is a great honor to receive your recognition.

This manuscript is a resubmission of an earlier submission. The following is a list of the peer review reports and author responses from that submission.
Round 1
Reviewer 1 Report
The authors of this paper proposed a conceptual framework for Blockchain E-bidding service. Although the topic of the paper is interesting and focuses on a practical problem, the paper suffers from several shortcomings.
1- The first and the most important problem is the English writing of this paper, which makes it unreadable and incomprehensible in some parts.
2- The abstract should be rewritten to show the importance of this paper and improve its fluency. Currently, it consists of many short sentences.
3- Motivation is not sufficiently stated in the introduction part. It should be clarified why the author considers this problem and what advantages of the proposed framework. The authors should revise the introduction and clarify the objective of this paper in detail precisely. A short description of the proposed method has been stated, which is not obvious to readers.
4- The structure is presented in Figure 2. (Underlying Architecture of Blockchain Technology) is incorrect in many parts. For example, EVM is not an application layer and Tendermint is not a consensus algorithm.
5- The paper lacks experiments or simulations.
Reviewer 2 Report
The authors propose an electronic bidding system for the construction/industry sector. The background and motivation have been explained well. However, several essential pointers are missing from this paper.
First, this paper has no proof-of-concept implementation and no evaluation of the proposed blockchain framework. Therefore, it is difficult to adjust the proposal's feasibility and prove the claim in the abstract "...to improve the overall 23 transaction process's security, stabilization, synergism and efficiency" (c.f., lines 21-23).
Second, if there is no implementation and evaluation, perhaps, this paper may be a concept paper. However, this paper still lacks many details, even for a concept paper. The explanations in Sections 3 and 4 are straightforward for readers with general backgrounds in blockchain applications. Therefore, no deep insights can be obtained after reading the current version of the manuscript.
Some other comments:
- Section 2.1.1 is unclear whether this section is the authors' proposed framework or a previously established framework from related works. If it is the authors' proposal, it should be moved from this section to the subsequent section.
- Change the title of Section 2.2.4 to represent the contents better. The current title, "Blockchain technology," is vague.
- Please use the journal's template for the next submission.
Reviewer 3 Report
This paper doesn't provide any technical novelty except some listing of applications of application. No mathematical analysis, no experimental analysis and not a single numerical result in the paper make invisible contributions by the authors. This paper needs thorough literature review to make author's contribution more visible. The paper needs thorough polishing for better presentation and typographical errors. The paper can't be accepted without a performance evaluation.
Reviewer 4 Report
The paper needs a proof-read. There are many grammatical errors and typos. For instance, in the abstract:
line 16: electronic bidding method demonstrate
line 20: du to its unique advantages,
Extend the contribution of this paper in section 1.
What is the contribution compared to related works?
Define blockchain pricesely in 2.2.1 [1]
[1] M. Raikwar, D. Gligoroski, and K. Kralevska, "SoK of Used Cryptography in Blockchain," IEEE Access, vol. 7, pp. 148 550–148 575, 2019
The paper doesn't include performance evaluation or proof of concept of the proposed system.